# Sexist textbooks: Automated analysis of gender bias in 1,255 books from 34 countries

**Lee Crawfurd**[1]*, **Christelle Saintis-Miller**[2], **Rory Todd**[1]

**1** Center for Global Development, London, United Kingdom, **2** Center for Global Development, Washington, DC, United States of America

* lcrawfurd@cgdev.org

## Abstract

Textbooks play a critical role in schooling around the world. Small sample studies show that many books continue to under-represent women and girls, and to portray men and women in stereotypical gendered roles. In this paper, we use quantitative text analysis to assess the degree of gender bias in a newly assembled corpus of 1,255 English language school text-books from 34 countries that are publicly available online. We find consistent patterns of under-representation of female characters and portrayal of stereotypical gendered roles. Women and girls appear less frequently, are portrayed as more passive, are less likely to be associated with work or achievement, and are more likely to be associated with the home and traditionally female occupations. Comparing across countries, female representation in books is correlated with higher GDP and more legal rights for women. Under-representation and stereotypes are a particular problem in South Asia.

**Data Availability Statement:** The code and data necessary to replicate all analysis can be accessed from the Harvard dataverse https://doi.org/10.7910/DVN/AJD0T4.

## Introduction

School textbooks play an important role in shaping identity, attitudes [1–5], and norms [6, 7]. Controversy over school book content is high in many countries including the US, where book bans continue today. The introduction of secular textbooks was critical in the political development of the religious right in America, and led to school boycotts, a shooting, and a bombing [8]. Despite the potential importance of books for shaping norms, still little is known quantitatively and systematically about the extent of biases in school books around the world.

In this paper we open the black box of global schooling, and provide quantitative insight into the content of school curricula and learning materials. Our particular focus is on girls education and gender bias. Girls education is a major global policy priority, but attention has focused primarily on the amount of schooling girls receive and the cognitive skills they obtain. If we care about equality of opportunity for boys and girls, we need to look beyond the building of human capital, to what children are exposed to in schools. Though there are many influences on children's gender norms, school books make up a large share of children's time spent at school, and may pass on implicit biases [9]. Many teachers make the majority of their instructional decisions based on textbooks [10].

**Funding:** This research was supported by funding from the Bill and Melinda Gates Foundation and Echidna Giving. The funders had no role in study design, data collection and analysis, decision to publish, or preparation of the manuscript.

**Competing interests:** The authors have declared that no competing interests exist.

Concretely, in this paper we ask how representation of males and females quantitatively differs across English-language textbooks globally, and to what extent textbooks portray stereotypes concerning the occupations, characteristics, and actions of male and female characters.

We use a newly compiled corpus of text documents, covering 1,255 school textbooks from 34 countries. We use 'textbooks' throughout as shorthand, though a small number of materials in our corpus are in fact student worksheets or slideshows. We process these documents using natural language processing and quantitative text analysis tools. We identify differences in the quantitative representation of male and female coded words, as well as in the qualitative descriptions of male and female words, using word co-occurrence, word embeddings, and dependency parsing.

Overall we find substantial gender bias in school textbooks across the 34 countries, with women and girls under-represented compared to men and boys, and portrayed in stereotypical feminine roles. In 28 of the 34 countries, women and girls occur less frequently than men and boys, and the gap for some countries is stark. Women tend to be portrayed in traditionally feminine occupations, such as nurse, teacher, and housekeeper. They are more likely than men to be associated with words relating to appearances and the home, compared to achievement and work. Women and girls are described as being more passive and less dominant than men, although they are also portrayed using more positive terms.

Comparisons between individual countries are limited by small samples, but higher income countries tend to have somewhat less unequal representation, although stereotypes still seem to be common. Countries with better representation also have better legal rights for women, even after adjusting for income levels.

Comparing across regions, South Asia has especially low representation of female characters, and somewhat stronger stereotypes, with more stereotypes in the adjectives used to describe male and female characters, a weaker association between female characters and work or achievement in general, and few women described as performing specific occupations.

Our analysis complements existing studies that have documented consistent gender bias in textbooks around the world, primarily through manual analysis and coding of texts [11, 12]. Studies on gender representation in textbooks from low- and middle-income countries come primarily from small-scale qualitative manual coding of books. Such studies typically show an under-representation of female characters, and associate women with lower-status occupations than men. This includes studies from Bangladesh [13], Cameroon [14], China [15], Ethiopia [16], Hong Kong [17], Iran [18], Nepal [19], Nigeria [20], Pakistan [21], and Palestine [22]. Some studies also present comparisons across a small number of countries, such as Bangladesh, Indonesia, Malaysia, and Pakistan [23], Australia, Hong Kong, Singapore, and Turkey [17, 24], and Rwanda, Kenya, and Uganda [25, 26].

A key advantage of our study is the ability to compare across a wider range of countries. Such larger comparative cross-country research is rare. One exception is a project at Stanford University looking at changes over decades, using books from the Georg Eckert Institute for International Textbook Research. One study reviews over 1,000 secondary school textbooks from 88 countries between 1950 and 2011, showing an increase in mentions of women's rights over time [27]. Another study manually codes 559 secondary school textbooks from 76 countries, focusing on representations of globalization [28]. Another uses a similar approach to look at long-term trends in human rights discussion in textbooks [29].

Methodologically, our study is similar to that of some others applying natural language processing to school textbooks from high-income countries. For example, one study analyzes the representation of race and gender in 1,000 award-winning American children's books, using both text and image analysis techniques [30]. A related study uses word embeddings in the same corpus, finding that women are more likely to be represented in relation to their

appearance than in relation to their competence, and are more likely to be represented in relation to their role in the family than their role in business [31]. A study of the 247 best-selling books aimed at children under 5 years of age in the United States and Canada found that many books include gender stereotypes, such as girls being better at reading and boys at math [32]. Another study uses word embeddings to show that in American history textbooks women are discussed more in the context of the home and less in the context of achievement [33]. More broadly, the use of quantitative text analysis to document gender bias is growing across a range of topics in economics and political science, including judge's opinions [34], Google News and Google Books [35, 36], local newspapers [37], and the Corpus of Historical American English [38].

Other studies on the impact of textbooks in developing countries have focused more on the provision of textbooks than on their specific content. For example, a randomized control trial of access to new books in Kenya found no impact on learning on average, and positive impacts only for the most able children [39]. An observational study using student fixed effects models found that textbooks had zero average effect on learning in 11 Anglophone African countries, but positive effects for children from the wealthiest families [40]. By contrast, packaged interventions including student books and teacher guides and training have been more successful, experimentally demonstrated in Kenya [41] and El Salvador [42].

In the remainder of this paper we first discuss the data and methods we use, then show our results, and finally discuss their significance.

## Materials and methods

### Data

**Textbook corpus.** Our data comes from a systematic review of online materials. From September 2020 to February 2022, we visited every ministry of education website in the world and downloaded all available educational materials (many countries made learning materials publicly available online to facilitate online learning during the Covid-19 pandemic). We manually reviewed the materials and selected only files that were in English and that could be classified as learning materials, rather than teacher guides or subject curricula. To ensure a broad range of countries were represented, we also searched for publicly available learning materials from websites other than ministries of education, focusing on countries with sizable populations and where English is an official language. Almost all of these books are published by government bodies, with the exception of the three high-income countries in our sample, where textbooks are published exclusively by the private sector (Australia, the United Kingdom, and the United States).

Our sample contains 1,255 textbooks from 34 countries, which total over 37.2 million words. These books cover a wide range of school subjects from grades 4 to 13 (see Tables 1 and 2). The included countries are: Afghanistan, Australia, Bangladesh, Belize, Bhutan, Dominica, Ethiopia, Guyana, India, Jamaica, Kenya, Kiribati, Lesotho, Liberia, Malawi, Maldives, Namibia, Nigeria, Pakistan, Papua New Guinea, Rwanda, Samoa, Sierra Leone, Solomon Islands, South Africa, South Sudan, Sri Lanka, St Kitts and Nevis, Tonga, Uganda, United Kingdom, United States, Zambia, and Zimbabwe (Fig 1). The countries in our sample, in which schooling is at least in part conducted in English, are of course different in one important way to other countries—many having a British colonial legacy. They are not necessarily so different in all relevant dimensions though. The countries cover a range of income levels and geographic regions, and have similar average levels of gender equality to other countries (see S1 Table).

**Table 1. Number of books, by country and grade.**

| | 1 | 2 | 3 | 4 | 5 | 6 | 7 | 8 | 9 | 10 | 11 | 12 | 13 | Total |
|---|---|---|---|---|---|---|---|---|---|---|---|---|---|---|
| Afghanistan | | | | 1 | 1 | 1 | 1 | 1 | 1 | 1 | 1 | 1 | | 9 |
| Australia | 10 | 8 | 7 | 4 | 6 | 7 | 11 | 7 | 11 | 7 | | | | 78 |
| Bangladesh | | | | 7 | 6 | 16 | 16 | 15 | | 26 | | | | 86 |
| Bhutan | | | | 8 | 2 | 1 | 1 | 1 | | 1 | 1 | 1 | | 16 |
| Dominica | | | | 2 | 2 | 1 | | | | | | | | 5 |
| Ethiopia | | 1 | 1 | | 1 | 2 | 4 | 1 | 7 | 6 | 1 | 9 | | 33 |
| Guyana | 11 | 5 | 5 | 5 | 4 | 4 | | | | | | | | 34 |
| India | 3 | 4 | 5 | 9 | 9 | 12 | 12 | 12 | 19 | 20 | 26 | 32 | | 163 |
| Kenya | 9 | 2 | 1 | | | | | | | | | | | 12 |
| Kiribati | | | | 3 | 7 | 7 | 6 | | 1 | | | | 1 | 25 |
| Malawi | 1 | 1 | 1 | 1 | | 1 | | | | | | | | 5 |
| Maldives | 10 | 8 | 7 | | | | | | | | | | | 25 |
| Namibia | | | | | | | | | | 5 | | 1 | | 6 |
| Nigeria | 9 | 2 | | | | | | | | | | | | 11 |
| Other | 2 | | | 5 | 1 | | | | 1 | 4 | 2 | 2 | | 17 |
| Pakistan | 3 | 3 | 3 | 7 | 7 | 9 | 8 | 10 | 16 | 10 | 10 | 8 | | 94 |
| Papua New Guinea | | | 3 | 3 | 3 | 3 | | | | | | | | 12 |
| Rwanda | 5 | 7 | 5 | 4 | 4 | 3 | 3 | 2 | 4 | 4 | 4 | 5 | | 50 |
| Samoa | | | | | | | | | 17 | 18 | 13 | 15 | 7 | 70 |
| Sierra Leone | | | | | | | 6 | 6 | 5 | 6 | 6 | 10 | | 39 |
| Solomon Islands | | | | 8 | 8 | 8 | 12 | 7 | 8 | 8 | | | | 59 |
| South Africa | | | | 7 | 6 | 6 | 9 | 8 | 6 | 3 | 2 | 2 | | 49 |
| South Sudan | 5 | 5 | 5 | 5 | 5 | 5 | 4 | 5 | 10 | 10 | 10 | 11 | | 80 |
| Sri Lanka | | | | 1 | 1 | 9 | 24 | 10 | 10 | 27 | 9 | 1 | | 92 |
| St Kitts and Nevis | | | | 4 | 4 | 4 | 7 | 6 | 1 | 13 | 9 | 6 | | 54 |
| Uganda | | | | | | | | 16 | 2 | 2 | 3 | 3 | 1 | 27 |
| United Kingdom | | 3 | 1 | 4 | 1 | 2 | 3 | 2 | 3 | | | 1 | | 20 |
| United States | 7 | 2 | 7 | 9 | 9 | 7 | 6 | 5 | 8 | 4 | 4 | 4 | | 72 |
| Zimbabwe | | 3 | | 3 | | | 2 | | 4 | | | | | 12 |
| Total | 75 | 54 | 51 | 98 | 87 | 109 | 136 | 114 | 134 | 175 | 101 | 112 | 9 | 1,255 |

Note: This table shows the number of textbooks in our corpus in each grade in each country. "Other" includes books from Belize, Jamaica, Lesotho, Liberia, Tonga, and Zambia—we treat these separately as there are fewer than 5 books from each individual country.

Although data on actual use of textbooks is scarce, the evidence available points to textbooks as being widely available and playing a dominant role in the curricular resources used in classroom instruction. Although there has long been concern about the scarcity of textbooks in poorer countries [43], most of the pupils in school surveys report access to at least a shared textbook. The 2019 Program for the Analysis of Educational Systems of CONFEMEN (PASEC) survey of 14 Francophone African countries (Benin, Burkina Faso, Burundi, Cameroon, Chad, Republic of the Congo, Democratic Republic of the Congo, Gabon, Guinea, Ivory Coast, Madagascar, Niger, Senegal, and Togo) found that on average 60–75 percent of 13-year-olds had their own textbooks [44]. The 2018 Programme for International Student Assessment (PISA) Development survey of 15-year-olds in seven countries (Cambodia, Ecuador, Guatemala, Honduras, Paraguay, Senegal, and Zambia) found that overall around 45 percent of students had their own textbook, and another 45 percent shared with others, with only 10 percent reporting no access at all [45]. The most recent 2012 Southern and Eastern Africa

**Table 2. Number of books, by country and subject.**

| | Ag | Ar | Ec | Ge | Ho | Hu | IT | La | Ma | PE | Re | Sc | So | Total |
|---|---|---|---|---|---|---|---|---|---|---|---|---|---|---|
| Afghanistan | | | | | | | | 9 | | | | | | 9 |
| Australia | | | | | | | | 46 | 28 | | | 4 | | 78 |
| Bangladesh | 4 | 4 | 4 | 9 | | 10 | 4 | 10 | 7 | 4 | 21 | 9 | | 86 |
| Bhutan | | | | | | | | 10 | 3 | 1 | | 1 | 1 | 16 |
| Dominica | | | | | | | | | 1 | | | 4 | | 5 |
| Ethiopia | | | 1 | 4 | | 2 | 3 | 10 | 3 | | | 10 | | 33 |
| Guyana | | | | 6 | | | | 6 | 15 | | | 7 | | 34 |
| India | | 3 | 8 | 3 | 2 | 21 | 2 | 32 | 34 | 1 | | 32 | 25 | 163 |
| Kenya | | | | | | | | 9 | 3 | | | | | 12 |
| Kiribati | | | 4 | | 1 | 3 | 1 | 3 | 5 | | | 8 | | 25 |
| Malawi | | | | 1 | | | | | 4 | | | | | 5 |
| Maldives | | 1 | | | | | 3 | 3 | 5 | 1 | 1 | 3 | 8 | 25 |
| Namibia | | 1 | | | | 2 | | 1 | | | | 2 | | 6 |
| Nigeria | | | | | | | | | 11 | | | | | 11 |
| Other | | | 1 | 5 | | | 2 | 3 | 4 | | | 2 | | 17 |
| Pakistan | | | | 9 | 2 | 3 | 10 | 20 | 12 | 1 | 8 | 27 | 2 | 94 |
| Papua New Guinea | | | | | | | | | 8 | | | 4 | | 12 |
| Rwanda | | | | | | 5 | | 8 | 9 | | 5 | 23 | | 50 |
| Samoa | 6 | 8 | 10 | | 8 | 3 | | 10 | 1 | 2 | | 14 | 8 | 70 |
| Sierra Leone | | | | | | | | 20 | 19 | | | | | 39 |
| Solomon Islands | | | | | | | | 32 | 24 | | | 3 | | 59 |
| South Africa | | | | 3 | | | 2 | 6 | 22 | | | 16 | | 49 |
| South Sudan | | | | 4 | | 8 | 4 | 12 | 14 | | 10 | 20 | 8 | 80 |
| Sri Lanka | | | 10 | 4 | | 15 | 9 | 23 | | 15 | 6 | 9 | 1 | 92 |
| St Kitts and Nevis | 1 | | | 9 | 9 | 6 | | 6 | 7 | | | 13 | 3 | 54 |
| Uganda | 1 | 3 | 1 | | 1 | 7 | 1 | 1 | 1 | 1 | 2 | 8 | | 27 |
| United Kingdom | | | 1 | 1 | | 5 | | 2 | 9 | | | 2 | | 20 |
| United States | | | 2 | 2 | | 5 | | | 47 | | | 16 | | 72 |
| Zimbabwe | | | | | | 2 | | 8 | 2 | | | | | 12 |
| Total | 12 | 19 | 43 | 60 | 14 | 100 | 47 | 352 | 236 | 26 | 53 | 237 | 56 | 1,255 |

Note: This table shows the number of textbooks in our corpus in each subject in each country. Ag = Agriculture, Ar = Arts and Music, Ec = Economics and Business, Ge = General Studies, Ho = Home Economics, Hu = Humanities, IT = Information Technology, La = Languages, Ma = Maths, PE = Physical Education, Re = Religion, Sc = Sciences, So = Social Science. "Other" includes books from Belize, Jamaica, Lesotho, Liberia, Tonga, and Zambia—we treat these separately as there are fewer than 5 books from each individual country.

Consortium for Monitoring Educational Quality (SACMEQ) survey of 14 Anglophone African countries, found that around 40 percent of grade 6 students owned reading and mathematics textbooks [46], with that number roughly doubled through sharing [40]. A 2015 Rwandan survey found that textbooks were widely available in all schools and in approximately three out of five classrooms [47].

Textbooks are also used as a primary instructional tool in the classroom, influencing not only what teachers teach, but how they teach [48]. The 2011 Trends in International Mathematics and Science Study (TIMSS) included survey questions on the types of materials used as the basis of instruction. 70 percent of fourth-grade science teachers and 74 percent of eighth-grade science teachers used textbooks the most often as the basis for instruction [49]. In mathematics instruction, 75 percent of fourth grade teachers and 77 percent of eighth grade

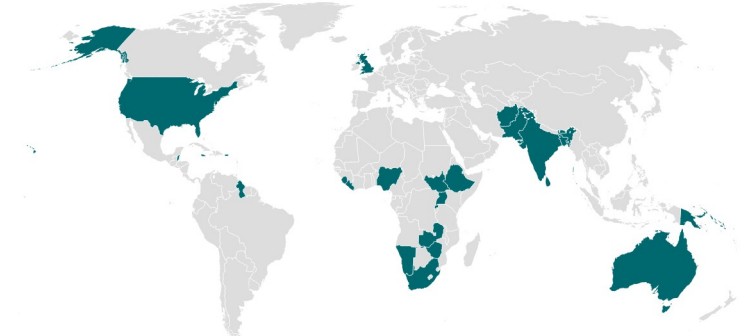

**Fig 1. Countries in corpus.** Note: Books from the 34 countries included in our corpus are coloured in teal. This includes Afghanistan, Australia, Bangladesh, Belize, Bhutan, Dominica, Ethiopia, Guyana, India, Jamaica, Kenya, Kiribati, Lesotho, Liberia, Malawi, Maldives, Namibia, Nigeria, Pakistan, Papua New Guinea, Rwanda, Samoa, Sierra Leone, Solomon Islands, South Africa, South Sudan, Sri Lanka, St Kitts and Nevis, Tonga, Uganda, United Kingdom, United States, Zambia, and Zimbabwe.

teachers used textbooks the most frequently. The World Bank Service Delivery Indicator (SDI) surveys conducted between 2012 and 2016 in 8 African countries found that 37 percent of children in randomly selected grade 4 classrooms had the relevant textbook [50] (this ranged from 88 percent in Morocco, through 68 percent in Togo, 66 percent in Mozambique, 43 percent in Kenya, 40 percent in Nigeria, 19 percent in Tanzania, 14 percent in Niger, to 11 percent in Madagascar). A study observing classrooms in successful reading programs found that they spent a large share of class time using textbooks (50 percent in Kenya, 45 percent in India, 42 percent in Senegal, 28 percent in Nigeria, and 8 percent in Tanzania) [51].

In the majority of countries in our sample, the government provides a set of specific approved textbooks directly to schools, rather than schools making their own choices. Therefore, particularly in these countries with centralized supply, we can expect large proportions of children in school to be actually exposed to the books in our corpus [52]. Typically, poorer countries have a single set of government-approved and provided books, whereas in richer countries, schools have more autonomy to select books from a variety of providers. Government-approved books are distributed to schools in 23 countries in our sample (Afghanistan, Bangladesh, Bhutan, Ethiopia, Kenya, Kiribati, Lesotho, Liberia, Maldives, Nigeria, Papua New Guinea, Rwanda, Samoa, Seychelles, Sierra Leone, Solomon Islands, South Sudan, Sri Lanka, Tonga, Trinidad and Tobago, Uganda, Zambia, and Zimbabwe). Schools have autonomy over textbook choice in Australia, Belize, Dominica, Guyana, Jamaica, Namibia, South Africa, St Kitts and Nevis, the United Kingdom, and the United States. India and Pakistan have a mixture of central government provision and local school choice in different states and provinces.

**Gendered word lists.** We compare the content of the textbooks in our corpus with predefined lists of gendered nouns, pronouns, and names. We adapt the gendered nouns and pronouns used in a previous study [30], including words such as she/her/woman and he/his/man. In total, there are 68 male words and 68 female words (listed in S1 File).

To identify names and their gender in our text, we use damegender [53], a tool for gender detection generated using census data from 20 countries. Given the international source of the textbooks, we find that this performs better than datasets using only names from one particular country, like the United States. We only allow names above a frequency of 10,000 in this dataset, since names with a frequency below this threshold were significantly less likely to be names in context. We exclude other names that are unlikely to be names in context, including country names, city names, and months of the year. We exclude names where the gender of

the individual may be ambiguous, defined as those where neither gender accounts for at least 95 percent of occurrences in the dataset.

We manually validated this process with a sample of 5 books on a range of subjects (English, Mathematics, Social Studies, Political Science, and Biology) and countries (Afghanistan, Bangladesh, Guyana, India, and Pakistan). We calculate both the rate of precision (the proportion of identified names that are in fact names in context) and recall (the proportion of names in the text which were identified). This validation process showed an average precision rate of 70 percent—meaning that from the names identified by the tool, 70 percent were actually names —and an average recall rate of 75 percent—meaning that the tool correctly identified 75 percent of the total names across the texts. We also examined the precision and recall rates between male and female names. The precision rate for female names was 65 percent compared to 78 percent for male names, meaning that the tool was able to identify male names that were actually names in the context of the text at a higher rate than female names. The tool was able to identify the number of female and male names at a similar rate, with the recall rate for female names 74 percent compared to 77 percent for male names. Because of this imprecision, we do not include names when measuring representation, instead only counting gendered nouns and pronouns. We do include names in the other parts of our analysis.

We have equal-length lists of 68 male nouns/pronouns and 68 female nouns/pronouns, but more unique female names (2,218) than male names (1,753).

**Family and work word lists.**   To construct lists of words in which we might expect gender bias, we adapt the lists used in several previous studies related to achievement, appearance, family, home, and work [31, 33, 54]. These include 13 words related to achievement, 6 related to appearance, 11 related to family and the home, and 20 related to work (full word lists are shown in S3 File).

To classify occupations we use a list of 1,156 occupation titles drawn from the Gazette (the official public record in the UK, and a commonly used source of text data, [55]). We then manually classify these occupations into three groups; professional, service, and manual occupations. We are left with 61 professional or managerial jobs, 23 service jobs, and 16 manual jobs (see S2 File).

**Country-related gender characteristics.**   We use several sources of data on gender norms and outcomes at the country level. First, the 2023 Social Institutions and Gender Index (SIGI) from the Organization for Economic Cooperation and Development (OECD). The SIGI index comprises 25 indicators covering discrimination, violence, bodily autonomy, economic rights, and civil liberties. Second, the World Bank Women, Business, and the Law Index measures women's economic and legal rights, and comprises 35 indicators. Third, the Center for Global Development (CGD) Girl's Education Policy Index (GEPI) measures policy effort related to girl's education across 32 indicators, including spending, sexual health, safety, employment, and role models [56]. Fourth, to measure equality in education *outcomes* we use the World Bank Gender Parity Index (GPI) in Gross Secondary Enrollment. Fifth, we look at data on teenage marriage of girls, and on female MPs, both drawn from the 2023 OECD Gender, Institutions, and Development (GID) Database. We standardize all variables to a mean of zero and a standard deviation of one.

## Methods

**Data processing.**   We begin by converting PDF books to text data, primarily using Optical Character Recognition (OCR) in Amazon Textract. This process uses a pre-trained model to infer text in an image, and recent advances in machine learning have greatly improved its accuracy; a recent benchmarking experiment found a word error rate of just 1.8% when

Textract was used to convert a clear image [57]. We partially validate the conversion process by manually comparing a sub-sample of text files to the original PDFs. For several countries where textbooks had machine-readable text, we used a function in R which uses the Poppler library of PDF tools. This tool is able to extract text directly and has comparable accuracy.

We group books by subject into 14 groups; Agriculture, Arts, Business and Economics, Environmental Studies, French, General and Social Studies, Home Economics, Humanities, IT, Language, Math, Physical Education, Science, and Social Science.

Due to the limited number of books available from each individual country, we show results by three country regions; these are (i) South Asia, (ii) Sub-Saharan Africa, (iii) the three high-income countries Australia, UK, and US. Other countries not within these regions are very heterogeneous, being distributed across the Caribbean and Pacific. We also code whether books are funded by a foreign development agency. We include all countries in which text-books include an acknowledgment of funding by a donor agency. This variable will include some measurement error if provided funding is not acknowledged.

**Measuring gender representation.**   We then measure the quantitative representation of male and female words in the corpus of textbooks. We do this by counting gendered nouns and pronouns. For each book, we calculate the share of female words as a percentage of all gendered words. We then show how the share of female words varies by country, subject, and grade. We also count the number of gendered words with a clear age classification (e.g., girl vs. woman and boy vs. man).

## Measuring gender stereotypes

We use three different methods to assess gender stereotypes. The simplest of these is simply counting word co-occurrences within sentences. Second we use word embeddings, which are numerical representations of words to quantitatively measure the similarity between gendered words and other thematic words. Third is a method known as 'dependency parsing', which automatically infers the relationships between words, allowing us to identify the specific verbs and adjectives which modify gendered nouns. Using a variety of methods increases our confidence in the results and aids understanding.

**Co-occurrence.**   The simplest way of measuring stereotypes is counting the number of co-occurences within a single sentence of gendered words and specific target words. We count all co-occurrences of job occupations with gendered terms. For this analysis, we assume that the male or female term refers to the individual with the occupation. Although this analysis is conceptually simple, this assumption may introduce some noise and bias. For example, the sentence "*he went to the doctor*" would count as a co-occurrence of a male term with the occupation doctor, though in this case the sentence does not indicate the gender of the doctor. We omit cases where both male and female terms are mentioned within the same sentence. We count cases of implicitly gendered occupation terms (such as 'salesman') as equivalent to a co-occurrence of a gendered term and a gender-neutral version of the term ('man' and 'salesperson').

**Word embedding.**   A more sophisticated approach to measuring stereotypes uses word embeddings. Word embeddings are numerical representations of words that capture their semantic and syntactic properties. We use the Word2vec algorithm [58], which encodes words as high-dimensional vectors or embeddings. Technically, Word2vec can be used to learn associations between any kind of token: for example, two-word phrases, or sentences. Word2vec trains a neural network on a corpus to perform one of two tasks: if implemented with a 'continuous bag-of-words model', to predict a token based on those surrounding it; and if implemented with a 'skip-gram' model, to predict the tokens surrounding a given token. Words

which are mathematically closer in this vector space tend to be used in similar contexts in the training corpus and can therefore be regarded as more closely associated. Embeddings can also be combined arithmetically. An often-cited example is that the vector resulting from *king − man + woman* will in a well-trained model be very close to the embedding for *queen*.

Word embeddings can be used for various natural language processing tasks, such as text classification, sentiment analysis, and machine translation. But they also reflect human-like biases that are present in the text data on which they are trained. For example, gender bias can be observed in word embeddings when certain words are more associated with male or female attributes or occupations. In our case, we are interested in the distance between male and female words on the one hand and words relating to work, family, achievement, and appearance on the other.

Word embeddings require a relatively large sample of words and sentences, and so we train our model on our entire corpus rather than on individual countries or sub-samples of the data. We first estimate the embedding for each individual word. Second, we calculate the cosine similarity between each gendered word and each target theme word. We do this only for gendered nouns and pronouns, rather than names: many names will be very low frequency and embeddings will be correspondingly imprecisely measured. The similarity can take values between -1 and +1. Identical vectors have a cosine similarity of 1, perpendicular vectors have a 90-degree angle between them and a cosine similarity of 0, and opposite vectors have an angle of 180 degrees between them and a cosine similarity of -1. Third, we take the average similarity between all male words and each target word, and all female words and each target word. Fourth, we calculate the difference between the female similarity and the male similarity for each target word. This leaves us with a measure of female bias for each individual word themed on home, work, achievement, or appearance. Fifth, to assess the uncertainty in this measure of female bias, we bootstrap standard errors [33]. We do this by splitting the entire corpus into sentences, sampling the sentences with replacement, and training a new model on this new sample. We repeat this process 50 times, and take the standard deviation of the pro-female bias estimates across the 50 models as the estimate of the standard error.

For our main set of results, we use a continuous bag-of-words model, with a window size of 6, and generate embeddings with 300 dimensions, following standard best practice [59]. We use negative sampling which reduces the computational cost of predicting embeddings by sampling a few negative examples (words that are not likely to appear in the context of the input word), and minimizing their probability. We calculate the similarity of embeddings as their cosine similarity.

**Dependency parsing.**   Dependency parsing is a method of syntactic analysis that determines the grammatical structure of sentences by identifying the relationships between words. Unlike word embeddings, which capture semantic relationships based on word co-occurrence patterns, dependency parsing focuses on recovering structural relationships within sentences. Specifically, it uses pre-trained models to infer the grammatical connections between words, providing insights into sentence structure and syntactic roles.

The 'universal dependencies' framework [60] underpins this approach by first performing 'part-of-speech tagging'—allocating all words in a language to one of a set of syntactic categories (such as adjective, noun, verb, etc.). This categorization depends both on the word itself and the context in which it is used. For instance, in English, 'butter' might be categorized as a noun or as a verb (as in 'to butter the bread'), depending on its usage in the sentence. We make use of the UDPipe model trained on the 'English Web Treebank' [61, 62], a corpus of 16,621 sentences sourced from the internet that have been manually annotated. Applying this model to our corpus, we first measure the frequency with which male and female terms appear as the *subject* rather than the *object* of sentences, which may suggest that either gender is

depicted in a more dominant role. We then identify the verbs and adjectives that are associated with male and female words, to explore how male and female characters are described, and what actions they are depicted doing.

Finally, in order to compare quantitatively the differences in how genders are portrayed across different country income levels, we conduct a sentiment analysis of the terms used with each gender. Specifically, we use the National Research Council Canada (NRC) Valence, Arousal, and Dominance (VAD) lexicon [63] which quantifies three aspects of words: valence —how pleasant or positive a word is; arousal—how active it is; and dominance—how dominant or submissive it is. This lexicon includes human ratings for more than 20,000 words. We calculate the average rating for each of the three dimensions for verbs and adjectives used with male and females terms, weighted by the frequency of the word.

### Cross-country correlations

Does the existence of bias in textbooks correlate with other measures of gender equality at the country level? We look at the cross-country correlations between bias in textbooks and various measures of gender equality. There are several important limitations to this analysis. First, we aren't able to make causal claims here, and causality could clearly run in both directions; from biased books to unequal norms, and vice versa from unequal norms to biased books. Nevertheless, we feel this is a useful test of the validity of the text analysis; a complete lack of correlation could raise doubts. Second, our sample of books is essentially a convenience sample and is not necessarily nationally representative of the books actually read by children in classrooms in each country. Third, we have a relatively small sample of countries at just 34, and so this analysis has low statistical power.

## Results

### Gender representation

Overall, across our full sample of books, there are more than twice as many occurrences of male words (178,142) as there are female words (82,113). There is also substantial variation between countries. After adjusting for book length, grade, and subject, the countries with the lowest representation of women and girls are Afghanistan, Pakistan, Sri Lanka, and South Sudan, where fewer than 1 in 3 gendered words are female (Fig 2). Inspecting the difference between subjects (across all countries), the lowest representations of female words are in religious studies and the humanities. High-paying subjects such as science and math also have less than equal representation (evidence shows relatively high labor market returns to studying math and science in India [64], the United States [65, 66] and Denmark [67].) The subject with the highest female representation is home economics. The representation of women and girls decreases with the length of the book, and there are no statistically significant differences in female representation across different grades. Among low- and middle-income countries, those with books funded by international aid projects have a higher, but still uneven female representation. All of this analysis is conducted with gendered nouns and pronouns and not with gendered names, due to the imprecision in the algorithm identifying names correctly in context. However the results are largely similar if we do include names—with the number of male words increasing to 232,502 and female words to 128,970. Representation by country only changes marginally (see S1 Fig). The share of overall variation explained by country and subject is around 8 percent each.

In general, we see a slightly more equal representation of young girls than adult women. Of the gendered words with a clear age classification (e.g., girl vs. woman and boy vs. man), female words make up 40 percent of child words, but only 36 percent of adult words (this is

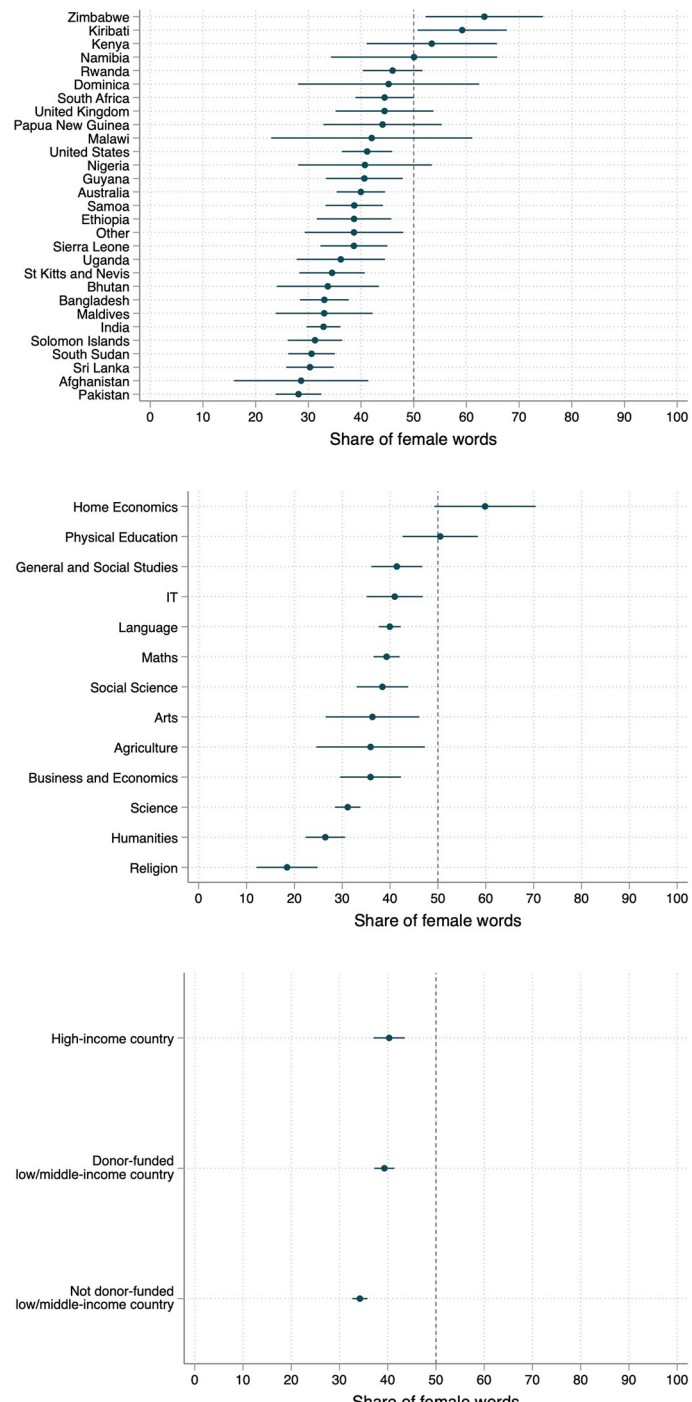

**Fig 2. Bias in gender representation, by country and subject.** Note: These figures show the predicted mean share of gendered words that are female. This measure is first calculated for each individual book, which are then estimated as a function of country, subject, grade, and (log) book length. We exclude countries with fewer than five books in our corpus. The high-income countries are Australia, the UK, and the United States. The low and middle income countries with donor-funded books are Bhutan, Ethiopia, Guyana, Kenya, Lesotho, Liberia, Namibia, Nigeria, Papua New Guinea, Samoa, Sierra Leone, South Africa, South Sudan, Tonga, Zambia, and Zimbabwe.

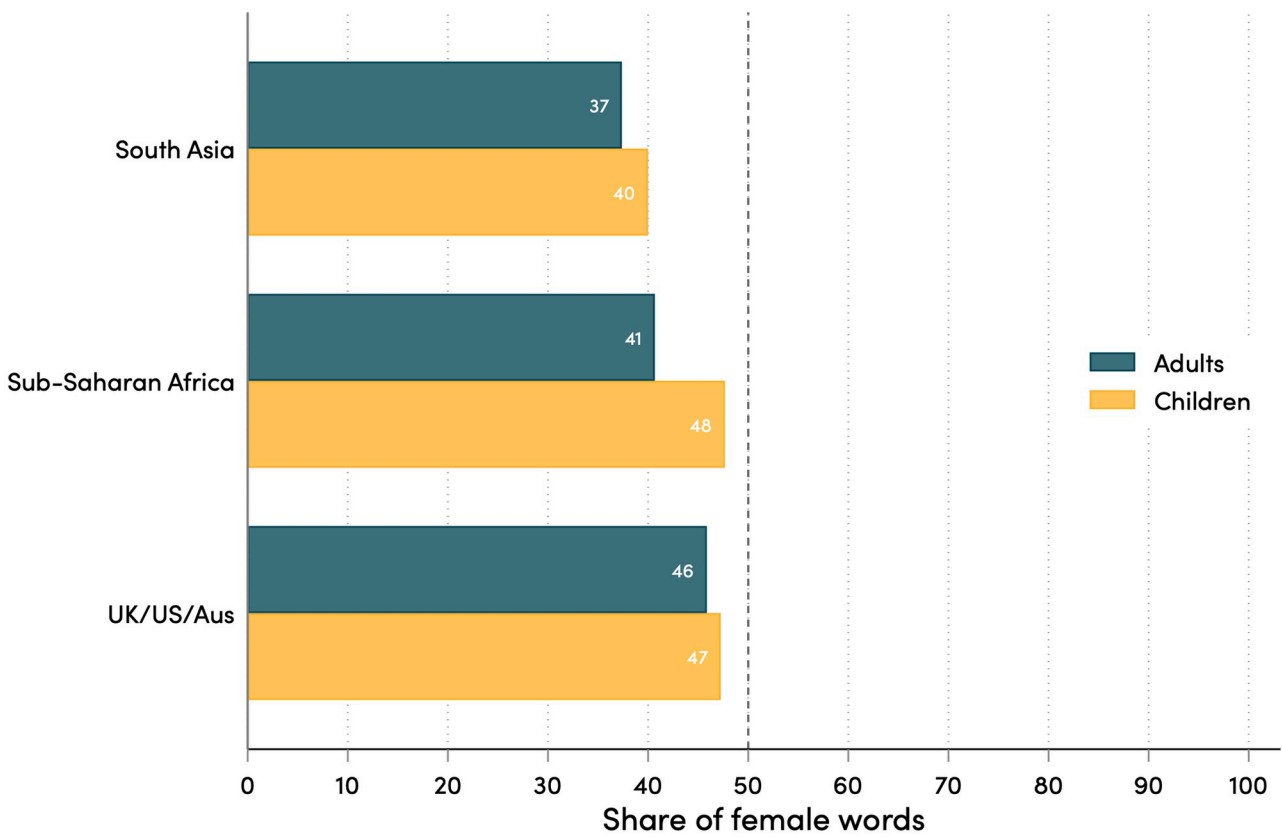

**Fig 3. Bias in gender representation, by age and region.** Note: These figures show the mean share of gendered words that are female. This measure is first calculated for each individual book.

similar to the pattern observed in the United States [30]). This gap is largest and only statistically significant for the sub-Saharan African countries (Fig 3).

## Gender stereotypes

**How women and girls are described.** To analyse stereotypes, we begin by identifying the adjectives and nouns used to describe women and girls in our corpus, using dependency parsing. Across our entire corpus our model identifies a total of 33,670 sentences containing an adjective and a gendered word, 202,287 sentences with a verb and a gendered word, and 164,626 sentences with a noun and a gendered word. In line with the general over-representation of male gendered words, we also see more co-occurrences of male words with specific adjectives and verbs. Dependency parsing shows no meaningful difference in the proportion of male and female words that appear as the subject rather than object of the sentence: 70.9 percent of female words and 70.3 percent of male words appear as the subject.

Focusing on words which occur relatively frequently in our corpus (at least five times each in the UK/US/AUS, South Asia, and Sub-Saharan Africa regions), we show the share of mentions alongside female terms as a proportion of all mentions alongside any gendered term. The adjectives most likely to describe female and not male characters include "married", "beautiful", "aged", and "quiet", and verbs include "bake", "cook", "shoe", and "sang". The adjectives mostly likely to describe male and not female characters include "powerful", "rich", "wise",

"certain", and "rich", although also "unable" (see S2 Fig), and verbs include "rule", "guide", "sign", and "order" (S3 Fig).

Exploring regional differences, we show these shares separately for the UK/US/AUS, South Asia, and Sub-Saharan Africa regions. In South Asia and Sub-Saharan Africa, "powerful" and "wise" have an even higher likelihood of being used to describe male characters compared to UK/US/AUS, although "rich" and "certain" are slightly less skewed toward male characters (Fig 4). Similarly, "aged" and "quiet" have a higher likelihood of being used to describe female characters in South Asia and Sub-Saharan Africa, although "married" and "beautiful" are slightly less skewed towards female characters. Exploring differences in verbs, "cook" is much less likely to be used to describe female characters in UK/US/AUS compared to South Asia and Sub-Saharan Africa (Fig 5).

In order to quantify the differences between words associated with male and female characters, we merge our annotated data with a set of quantified scores for the 'Valence', 'Arousal', and 'Dominance' of words (these roughly describe the positivity, activity, and power associated with a word, respectively). In general, we find that female characters are associated with more positive words (valence, +0.07 standard deviations), and male characters with more active (arousal, +0.04 standard deviation difference) and dominant words (+0.08 standard deviations). These averages mask substantial heterogeneity across countries (S5 Fig). Nigeria stands out as having particularly large differences (in favour of men) in all three scores.

**Association of gendered words with stereotypes around achievement, appearance, work, and family.** In general, we find that, in line with traditional stereotypes, female words are more likely to be associated with home- or family-related words and with appearance-related words. Male words are more likely to be associated with work- and achievement-related words. The words with the strongest male bias are "leader", "authority", and "powerful". The words with the strongest female bias are "wedding", "slim", and "cousins" (Fig 6). Almost all of the individual achievement and work-themed words have a stronger association with male words than female words. All individual appearance and home-themed words have a stronger association with female words than male words.

To assess heterogeneity, we consider the differences between regions, and between country income groups (we do not look individually at all countries due to advice that word embeddings models are run only on a corpus with at least 1.5 million 'tokens' or words [34]. Only ten countries in our sample have this many tokens.) South Asia shows a stronger male bias for terms associated with 'achievement', and both South Asia and Sub-Saharan Africa show a stronger female-bias for 'appearance' terms, than UK/US/AUS (Fig 7). Sub-Saharan African also shows a stronger female bias for terms relating to the 'home', although the region shows less male bias towards 'work' terms, compared to UK/US/AUS and especially South Asia.

We pool together countries into four income groupings following the World Bank classification; low-income, lower-middle income, upper-middle income, and high-income. Overall we see similar levels of pro-male bias on average in similarity to achievement-themed and work-themed words (S6 Fig). This pro-male bias is smaller though in the upper-middle income countries in our sample (Belize, Dominica, Jamaica, Namibia, South Africa, and Tonga). All country groups have a pro-female bias in similarity with appearance- and home-themed words. This pro-female bias is largest in the books from the low-income countries in our sample (Afghanistan, Ethiopia, Rwanda, South Sudan, Liberia, Sierra Leone, and Uganda).

In S7 Fig, we also show embedding biases for the ten countries in our sample with more than 1.5 million tokens (recommended as a minimum corpus size for estimating reliable embeddings [34]). Patterns are difficult to discern, however, likely due to the still relatively small size of the corpora.

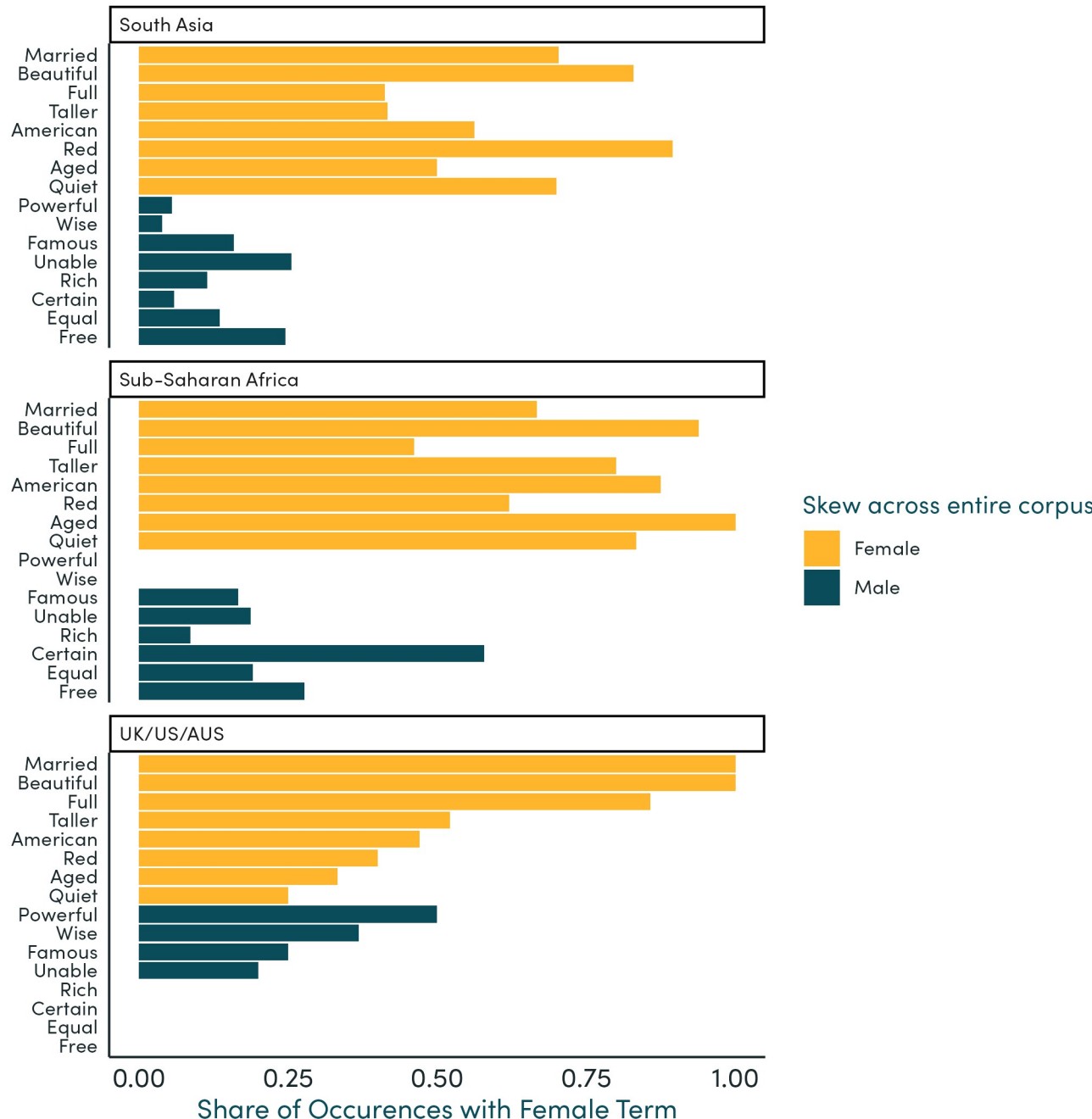

**Fig 4. Adjectives most likely to be used for each gender, by region.** Note: This figure shows the share of occurrences of adjectives used to describe people, which were used for female terms or nouns. We show here the 12 adjectives which were the most skewed toward either gender across UK/US/ AUS, South Asia, and Sub-Saharan Africa, provided that they occurred at least 5 times in textbooks for each region. Shares are shown separately for each region.

**Gendered occupations.** Overall, a clear majority of job occupations co-occur with male rather than female nouns and pronouns. Of the 20 most frequently mentioned occupations, all but 'domestic help' and 'nurse' co-occur more with male than female nouns and pronouns (S9 Fig). In relative terms, among occupations which appear at least ten times across our corpus, those that occur most frequently alongside female words include 'nurse', 'domestic

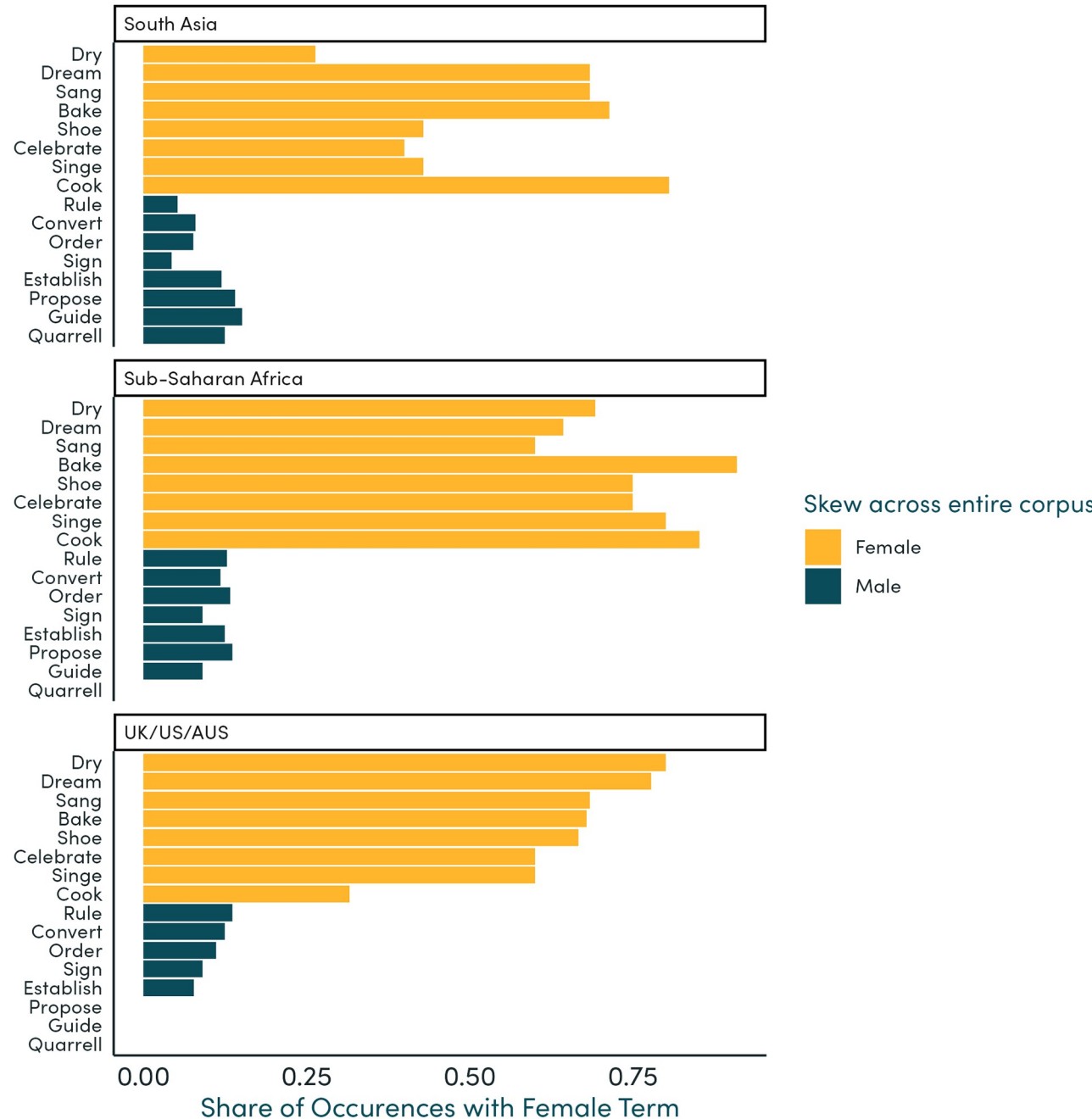

**Fig 5. Verbs most likely to be used for each gender, by region.** Note: This figure shows the share of occurrences of verbs used to describe people, which were used for female terms or nouns. We show here the 12 verbs—to be precise, the lemmas of verbs—which were the most skewed toward either gender across UK/US/AUS, South Asia, and Sub-Saharan Africa, provided that they occurred at least 5 times in textbooks for each region. Shares are shown separately for each region.

helper', and 'religious official' (see S10 Fig). The most male-dominated occupations include 'physicist', 'mathematician', and 'salesperson'.

Across country groups, we see a particularly high share of male-bias in reference to common occupations in South Asia and Sub-Saharan Africa (Fig 8). In the high-income countries (Australia, UK, and US), 6 of the most common 15 occupations have a female-bias or

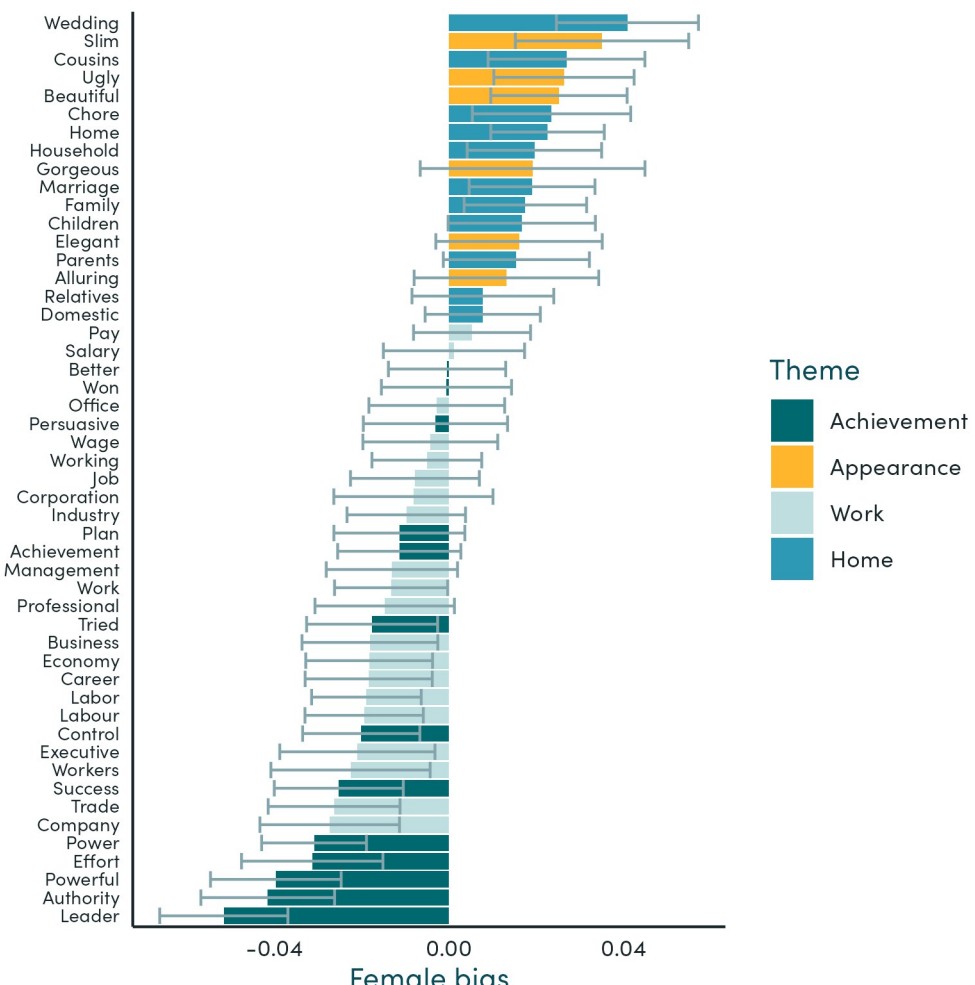

**Fig 6. Similarity of gendered words to stereotypical themes.** Note: This figure shows how terms related to four themes are associated with gender terms in our embeddings. Female bias is calculated as the difference between the average cosine similarity of the theme word with the set of female gender terms and the average similarity of the theme word with the set of male gender terms. The confidence intervals are calculated as the standard deviation for this statistic, over 50 bootstrap samples, where samples are generated by sampling all sentences in our corpus with replacement.

approximately even gender split. By contrast 14 of the most common 15 occupations have a male-bias in South Asia. Amongst these most common occupations, those with the highest share of female mentions include 'domestic help', 'teacher', and 'student'.

Amongst those occupations that do co-occur with female nouns and pronouns, they are slightly more likely to be managerial and professional roles, and less likely to be manual jobs, compared to those co-occurring with male nouns and pronouns (S11 and S12 Figs). This is not the case for South Asia, however (Fig 9). In general, occupations in UK/US/AUS are more likely to be managerial and professional roles, compared to other regions (Fig 9).

## Country gender laws, norms, and textbooks

To what extent does gender bias in textbooks reflect gender-related social norms? We find a strong correlation between our measure of gender representation in textbooks and other

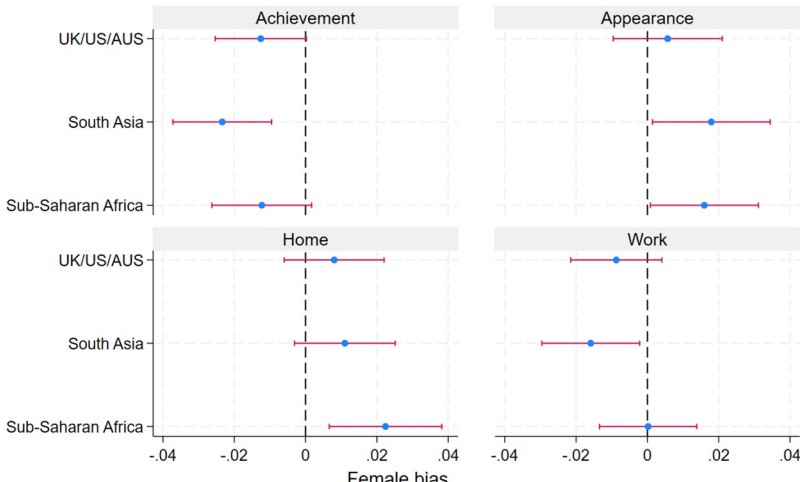

**Fig 7. Similarity of gendered words to stereotypical themes, by region.** Note: Female bias is calculated as the difference between the average cosine similarity of the theme word with the set of female gender terms and the average similarity of the theme word with the set of male gender terms. Confidence intervals are calculated as the standard deviation for this statistic, over 50 bootstrap samples, where samples are generated by sampling all sentences in our corpus with replacement, until a sample has as many sentences as the original corpus.

measures of gender bias at the country level. A 1 standard deviation increase in the SIGI index, WBL index, or GEPI index is associated with a 5–6 percentage point increase in the share of gendered words in books that are female. A 1 standard deviation increase in the gender parity in secondary schooling index, or in the share of female members of parliament, is associated with a 2 percentage point increase in the share of female words in books. Increase in rates of girl marriage are associated with lower shares of female words in books. All of these correlations are estimated in regressions in which the outcome is the share of gendered words in a specific textbook that are female, estimated as a function of country-level attitudes, outcomes, and laws, whilst controlling for (log) GDP per capita, and characteristics of the textbook itself (its grade, subject, and word length) (Table 3).

## Discussion

How much does bias in textbooks really matter?

Written materials, including textbooks, curricula, and exams, play a major role in schooling. Thus, how textbooks show girls and boys, women and men, in various different roles, and the portrayal of males and females might influence how children and adolescents view gender roles in society. But how important are they?

The main theory of gender development posits that environmental factors interact with three major social modes of influence; (i) modeling by people in one's immediate environment and through mass media, (ii) through social reactions to one's own behavior, and (iii) through direct tuition. All of these factors are expected to interact in complex ways, but with the largest influence coming from modeling rather than direct tuition [68]. We also know that children's beliefs about gender stereotypes start young. One study showed that boys and girls as young as six years old thought that boys were more likely to be 'brilliant' than girls [69].

Evidence on the causal impact of bias in textbooks on children is limited. One exception is a study from Bangladesh that showed that providing children with gender-sensitive stories made them more likely to agree that boys and girls should do the same things when they grow up, both do chores, and both play football [70]. Small-scale lab experiments in the United

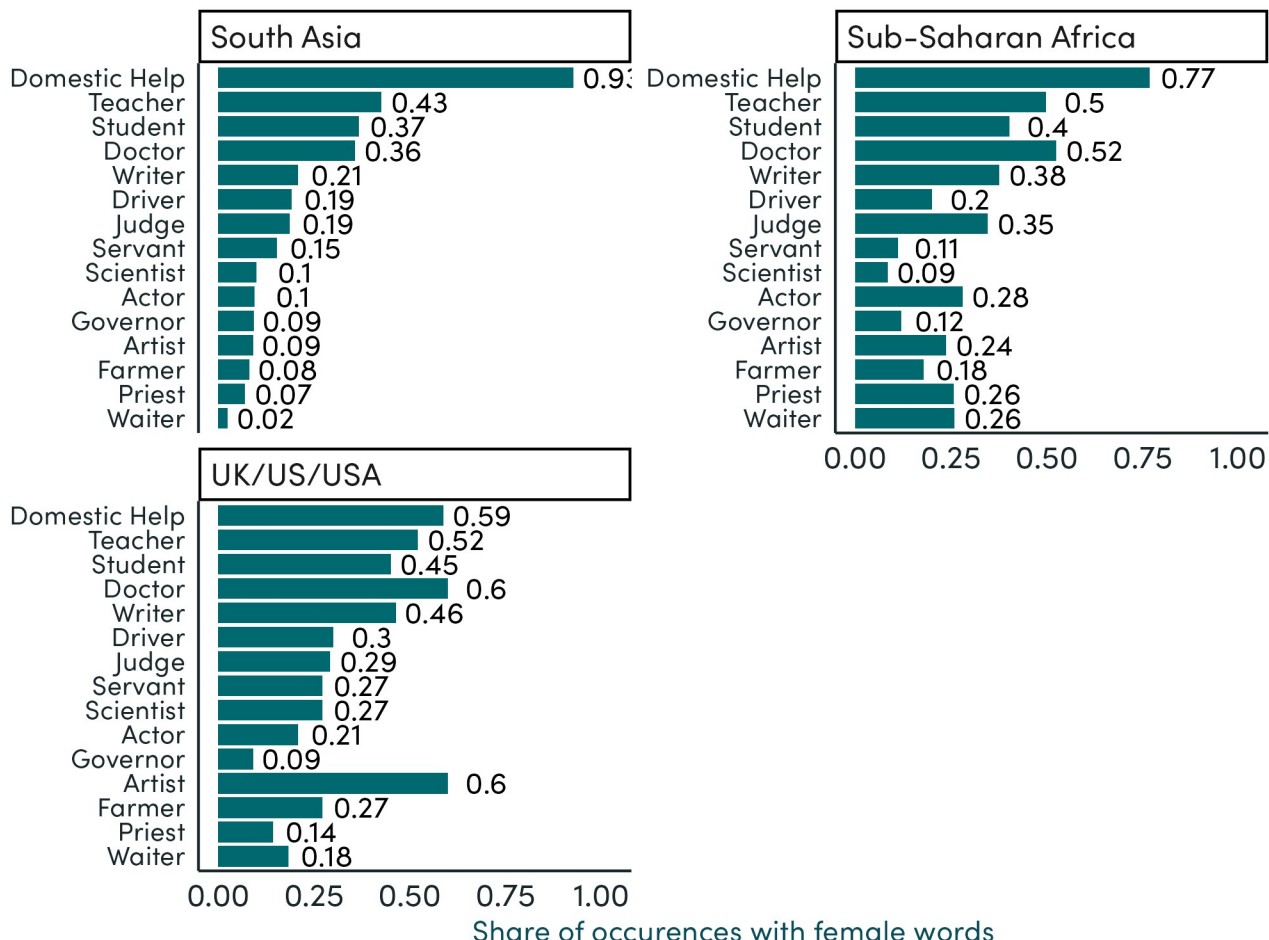

**Fig 8. Co-occurrence of job occupations with gendered terms (percent female), by region.** Note: This figure lists the 15 most common occupations in our full text corpus. The x-axis shows the share of all co-occurrences of the occupation and a gendered word which are female. Shares are shown separately for South Asia, Sub-Saharan Africa, and UK/US/USA.

States have shown that children express a preference for same-gender characters in books, and their behavior is more influenced by the behaviour of same-gender characters in storybooks [71, 72]. Children taught with gender-stereotyped books are more likely to express gender-stereotyped views on appropriate activities, personality characteristics, and career roles [73, 74]. Experimental evidence also shows that gender stereotypes can be reduced by exposure to counterexamples [75, 76].

A growing literature in economics documents the importance of role models and aspirations in determining schooling and employment outcomes, which can plausibly be provided by fictional characters as much as in-person interactions [77, 78]. Experimental evidence from Australia found that showing books with role models of the same sex to grade 3 children increased self-esteem, both for boys and girls [78]. This notion is also supported by Kenyan survey data showing that children were more likely to empathize with characters in books of their own gender [79].

Outside of the classroom, gendered language in school exams has been shown to lead to poorer performance for girls [80]. Countries with gendered national languages have worse employment and education outcomes for women [81].

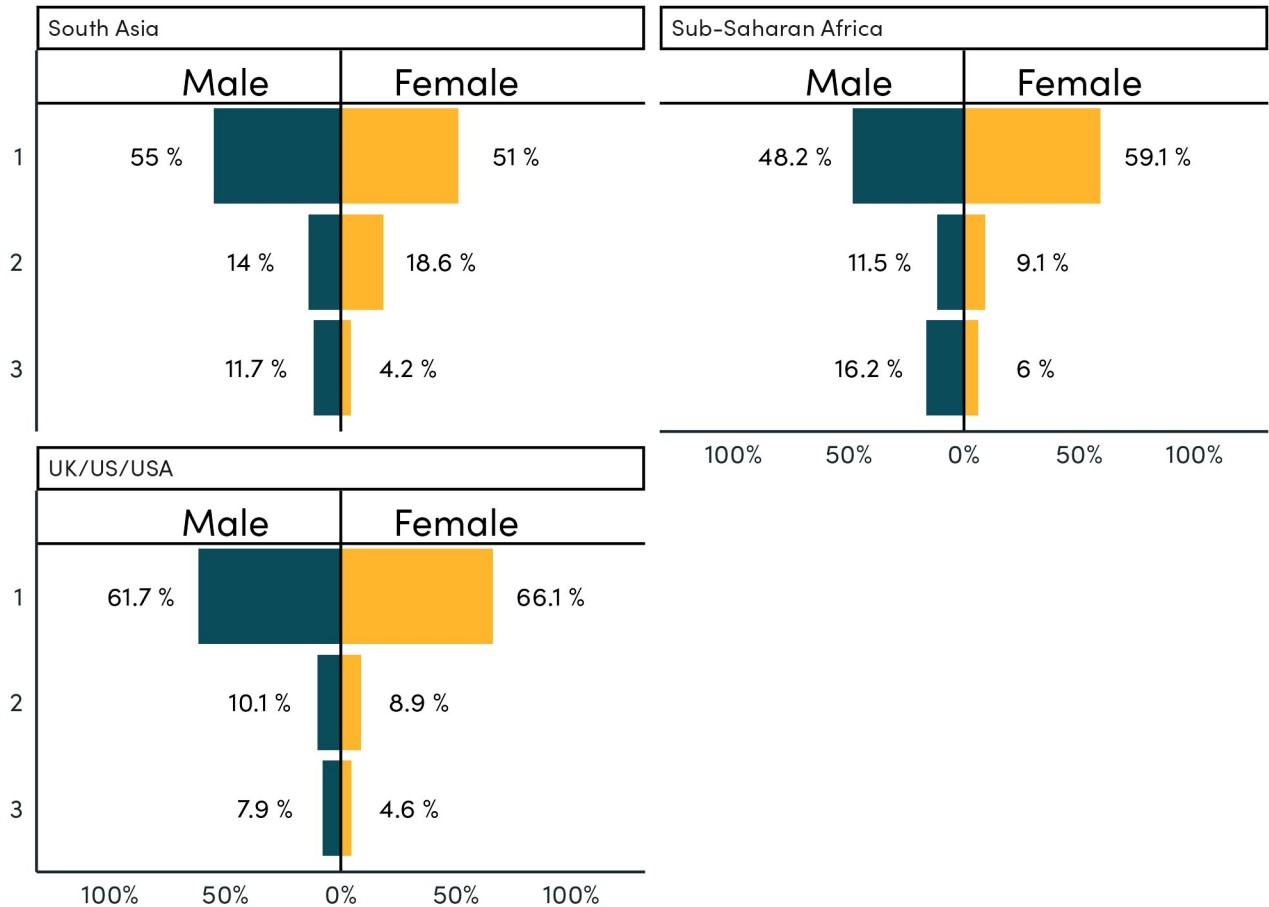

**Fig 9. Co-occurrence of job occupations with gendered terms, by job type and region.** Note: This figure shows the distribution of occupation categories across co-occurrences with gendered words. That is, for all co-occurrences between a male gendered word and an occupation for textbooks from South Asia, 66 percent of those co-occurrences were with managerial or professional occupations (category 1). 15 percent with service (category 2) occupations, and 18.9 percent with manual (category 3) occupations.

## Conclusion

In this paper, we gather a new corpus of 1,255 textbooks from 34 countries, and document under-representation of female characters and gendered stereotypes. Textbooks from low- and lower-middle-income countries show a greater degree of bias than upper-middle- and high-income countries. Lower female representation in books correlates with other measures of gender equality.

There are several limitations to our analysis. First, we focus only on text and not images. Advances in computer image recognition allow for the identification of different characters in pictures, though this is beyond the scope of our paper. Second, we focus exclusively on English language books, though our approach is in principle compatible with other languages. Third, while we manually validate the automated processes we use where possible, several of our analytical methods entail some imprecision in measuring bias. For example, some 'names' identified may not be names in context, and the co-occurrence of a gender term with an occupation term will not always indicate they are connected. Fourth, we are not able to distinguish sections of the text that are intended to be read by students from sections that are not, such as information about the book publisher. Fifth, while we recognize that gender can be expressed

**Table 3. Country characteristics and bias in textbook gender representation.**

| | (1) share | (2) share | (3) share | (4) share | (5) share | (6) share | (7) share |
|---|---|---|---|---|---|---|---|
| SIGI Index (z-score) | 5.655*** | | | | | | 4.190 |
| | (0.839) | | | | | | (3.514) |
| WBL Index (z-score) | | 4.965*** | | | | | 2.900 |
| | | (1.300) | | | | | (3.755) |
| GEPI Index (z-score) | | | 4.009*** | | | | 0.264 |
| | | | (1.179) | | | | (1.426) |
| Sec Ed Parity (z-score) | | | | 2.109** | | | 2.366** |
| | | | | (0.897) | | | (0.906) |
| Girl marriage (z-score) | | | | | -2.176** | | 0.812 |
| | | | | | (0.997) | | (2.057) |
| Female MPs (z-score) | | | | | | 2.971*** | -0.573 |
| | | | | | | (0.732) | (1.112) |
| Log GDP | -1.387* | -1.284 | -0.755 | 0.464 | 0.229 | 0.625 | -2.077** |
| | (0.686) | (0.793) | (0.858) | (0.557) | (0.651) | (0.628) | (0.735) |
| Controls | Yes | Yes | Yes | Yes | Yes | Yes | Yes |
| Obs. (Books) | 919 | 1,191 | 1,141 | 1,191 | 1,121 | 1,112 | 919 |
| Obs. (Countries) | 20 | 29 | 27 | 29 | 27 | 26 | 20 |
| $R^2$ | 0.216 | 0.158 | 0.157 | 0.134 | 0.145 | 0.155 | 0.227 |

Note: The outcome variable is the share of gendered words in each textbook that are female. This value can range between 0 and 100. Controls include the log of GDP per capita, as well as individual book characteristics—their subject, grade level, and word length. The SIGI Index is the OECD's Social Institutions and Gender Index. The WBL Index is the World Bank's Women, Business, and the Law Index. The GEPI Index is CGD's Girl's Education Policy Index. Data on Secondary Education Parity is from UNESCO. Data on girl marriage and female MPs is from the OECD's Gender, Institutions, and Development database.

* $p < 0.1$,

** $p < 0.05$,

*** $p < 0.01$. Standard errors in parentheses.

and perceived in a variety of ways, in this analysis we use a binary view of gender. Part of our reasoning for this more narrow view lies in the books included in our corpus, which only include binary gendered characters. We therefore use the same view in our analysis, though the absence of non-binary characters and adoption of a binary view of gender does reveal something about the norms around gender that are perpetuated by the countries included in our analysis.

Our analysis highlights which countries and subjects have particularly low female representation. This presents a new front in the agenda for girls' education policy. Several countries in our database have achieved gender parity in school enrollment, but still see significant underrepresentation of women and girls in school materials. Paying particular attention to gender representation could be prioritized in future revisions of these books. A constraint to such reform is the likelihood that reform is most needed in countries in which it is least likely due to regressive gender norms amongst adults. This may be addressed in part in countries in which government textbooks are partly funded by donors.

In addition to highlighting the scope for revision of existing books, our analysis can also be easily replicated with new books, and so could be used as a proactive check for new drafts.

Our analysis was limited by the public availability of books, even when they were publicly funded by governments with policies in place mandating open access to materials. Donors and national governments should ensure that all publicly funded school books are easily available

online in digital copy. Digital books are public goods, which should be freely available. Book availability to households is particularly important in the context of increasingly frequent school closures due to climate-related shocks, as well as shocks related to conflict and disease outbreaks. The opening of schoolbooks to researchers is just an added benefit.

Future research could easily expand our analysis to more datasets. An advantage of the quantitative approach is that our analysis is easily scalable and can be extended to new corpora of digitized books as they become available. New analyses of image data would also be valuable. As our analysis is largely descriptive, new experimental research would be useful in documenting the impact of gender bias in books on student attitudes. Replacing textbooks with more progressive ones may not be enough to change attitudes if teachers do not engage with them [82, 83]. But while debiasing school textbooks may not be sufficient to change attitudes, the clear bias demonstrated by our analysis suggests that it may be an important first step.

## Supporting information

**S1 Fig. Bias in gender representation, by country, subject, and income group (including names).** Note: These figures show the predicted mean share of gendered words that are female. Here, word counts for each gender include names identified as male or female. This measure is first calculated for each individual book, which are then estimated as a function of country, subject, grade, and (log) book length. We exclude countries with fewer than five books in our corpus. The high-income countries are Australia, the United Kingdom, and the United States. The low and middle income countries with donor-funded books are Bhutan, Ethiopia, Guyana, Kenya, Lesotho, Liberia, Namibia, Nigeria, Papua New Guinea, Samoa, Sierra Leone, South Africa, South Sudan, Tonga, Zambia, and Zimbabwe.
(ZIP)

**S2 Fig. Adjectives most likely to be used for each gender (full corpus).** Note: This figure shows the share of occurrences of adjectives used to describe people, which were used for female terms or nouns. We show here the 12 adjectives which were the most skewed toward either gender across UK/US/AUS, South Asia, and Sub-Saharan Africa, provided that they occurred at least 5 times in textbooks for each region.
(TIF)

**S3 Fig. Verbs most likely to be used for each gender (full corpus).** Note: This figure shows the share of occurrences of verbs used to describe people, which were used for female terms or nouns. We show here the 12 verbs—to be precise, the lemmas of verbs—which were the most skewed toward either gender across UK/US/AUS, South Asia, and Sub-Saharan Africa, provided that they occurred at least 5 times in textbooks for each region.
(TIF)

**S4 Fig. Verbs, adjectives, and nouns used alongside gendered terms (dependency parsing).** Note: These figures show the counts of the number of occurrences of gendered words alongside different specific adjectives, verbs, and nouns. The gendered words used are those contained in S1 File and the names described in Section. We identify adjectives, verbs, and nouns using the dependency parsing approach described in Section.
(TIF)

**S5 Fig. Valence, Arousal, and Dominance scores for adjectives and verbs used alongside gendered terms, by country.** Note: These figures show the differences in average scores (in Valence, Arousal, and Dominance) for adjectives and verbs used for female compared to male

gender terms. The difference is shown by country.
(TIF)

**S6 Fig. Gendered word associations, by country income group.** Note: Female bias is calculated as the difference between the average cosine similarity of the theme word with the set of female gender terms and the average similarity of the theme word with the set of male gender terms. Confidence intervals are calculated as the standard deviation for this statistic, over 50 bootstrap samples, where samples are generated by sampling all sentences in our corpus with replacement, until a sample has as many sentences as the original corpus. H indicates high-income countries, which are Australia, Canada, the United Kingdom, the United States, Guyana, and St Kitts and Nevis. UM are upper-middle-income countries which are Belize, Dominica, Jamaica, Namibia, South Africa, and Tonga. LM are lower-middle-income countries, which are Bangladesh, India, Pakistan, Kenya, Nigeria, Zambia, Zimbabwe, Bhutan, Kiribati, Papua New Guinea, Samoa, Solomon Islands, Sri Lanka, and Lesotho. L are low-income countries, which are Afghanistan, Ethiopia, Rwanda, South Sudan, Liberia, Sierra Leone, and Uganda.
(TIF)

**S7 Fig. Gendered word associations, for countries with at least 1.5 million tokens.** Note: Biases are shown only for the 10 countries in our corpus with at least 1.5 million tokens, as embeddings generated for other countries will be estimated unreliably. Female bias is calculated as the difference between the average cosine similarity of the theme word with the set of female gender terms, and the average similarity of the theme word with the set of male gender terms. Confidence intervals are calculated as the standard deviation for this statistic, over 50 bootstrap samples, where samples are generated by sampling all sentences in our corpus with replacement, until a sample has as many sentences as the original corpus.
(TIF)

**S8 Fig. Gendered word associations.** Note: Words above the 45˚ line are discussed more frequently in relation to women and girls, words below the line more frequently in relation to men and boys.
(TIF)

**S9 Fig. Co-occurrence of job occupations with gendered terms (percent female) (full corpus).** Note: This figure lists the 15 most common occupations in our full text corpus. The x-axis shows the share of all co-occurrences of the occupation and a gendered word which are female.
(TIF)

**S10 Fig. Co-occurrence of job occupations with gendered terms (percent female), most biased jobs.** Note: This figure shows common occupation terms—classified as those with more than 10 occurrences across the corpus—which were most biased towards either gender, in the sense of co-occurring (within sentences) more frequently with terms of one than the other.
(TIF)

**S11 Fig. Co-occurrence of job occupations with gendered terms, by job type (percent).** Note: This figure shows the distribution of occupation categories across co-occurrences with gendered words. That is, for all co-occurrences between a male gendered word and an occupation, 64.8 percent of those co-occurrences were with managerial or professional occupations (category 1). 13.7 percent with service (category 2) occupations, and 21.4 percent with manual

(category 3) occupations.
(TIF)

**S12 Fig. Co-occurrence of job occupations with gendered terms, by job type (absolute counts).** Note: This figure shows the distribution of occupation categories across co-occurrences with gendered words. In this case, absolute values are shown. For example, there were 3204 co-occurrences between male gendered words with a managerial or professional (category 1) occupation.
(TIF)

**S13 Fig. Co-occurrence of job occupations with gendered terms, by job type and country.** Note: This shows the results described above for the nine countries with over 250 occupation term-gender term cooccurences.
(TIF)

**S14 Fig. Gender representation by country, and country-level indicators of gender equality.** Note: This figure shows correlations between gendered word representation and country level indicators of gender equality. SIGI is the OECD Social Institutions & Gender Index. WBL is the WB Women, Business, & the Law Index. GEPI is CGD's Girl's Education Policy Index. Secondary Education Parity is from UNESCO. Girl marriage and female MPs are from the OECD Gender, Institutions, & Development database.
(TIF)

**S15 Fig. Gender representation by country, and country-level indicators of gender equality.** Note: This figure shows correlations between gendered word representation and country level indicators of gender equality. Support for wife-beating is from the Demographic & Health Survey. Female labour force participation is from the WB World Development Indicators. Support for basic rights and attitudes to women working are from the Gallup World Poll.
(TIF)

**S1 File. Gendered word lists.** A list of all male and female gendered nouns and pronouns for which we search.
(DOCX)

**S2 File. Occupation job type word list.** A list of all occupations for which we search.
(DOCX)

**S3 File. Theme word lists.** A list of thematic words related to 'home', 'achievement', 'appearance', and 'work'.
(DOCX)

**S4 File. Sample textbook page from Sierra Leone, grade 10 English.** Note: In this sample page from a textbook we can see an uneven representation of male and female nouns and pronouns—there are 6 male themed words (he/his) and 4 female themed words (ladies/she/her/ladys).
(TIF)

**S5 File. Sample textbook page from Pakistan, grade 5 Social Studies.** Note: In this sample page we see the Prime Minister being discussed as He/him/his. The current Prime Minister does happen to be a man, but he is not named in this book, and the role is clearly not inherently gendered.
(TIF)

**S6 File. Sample textbook page from Pakistan, grade 7 Home Economics.** Note: This book discusses how "you can help your mother" with housework.
(TIF)

**S7 File. Example sentences of gender stereotypes.**
(TIFF)

**S1 Table. Gender equality in countries in our sample vs rest of the world.**
(DOCX)

**S2 Table. Publishing information for textbooks from each country.** Note: This table shows the sources and publishing information for the textbooks included in our corpus.
(DOCX)

# Acknowledgments

We are grateful for research assistance to Laura Moscoviz, Emma Cameron, and Theo Mitchell, for helpful comments and advice to Anjali Adukia, Susannah Hares, Alice Evans, Margaret Leighton, David Evans, Asyia Kazmi, Izzy Boggild-Jones, Shubha Jayaram, and two anonymous referees, and for help in sourcing textbooks to Ezioma Akabike and Mo Olateju.

# Author Contributions

**Conceptualization:** Lee Crawfurd, Christelle Saintis-Miller, Rory Todd.

**Data curation:** Lee Crawfurd, Christelle Saintis-Miller, Rory Todd.

**Formal analysis:** Lee Crawfurd, Christelle Saintis-Miller, Rory Todd.

**Writing – original draft:** Lee Crawfurd, Christelle Saintis-Miller, Rory Todd.

**Writing – review & editing:** Lee Crawfurd, Christelle Saintis-Miller, Rory Todd.

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
