## [Decision Letter · Decision Letter 0]

11 Mar 2024

PONE-D-24-04364Sexist TextbooksPLOS ONE

Dear Dr. Crawfurd,

Thank you for submitting your manuscript to PLOS ONE. After careful consideration, we feel that it has merit but does not fully meet PLOS ONE’s publication criteria as it currently stands. Therefore, we invite you to submit a revised version of the manuscript that addresses the points raised during the review process.

**You will see from the reviewers' reports that they have a very positive view of your paper, and that chimes with my own view. Reviewer 1's concerns about some of your methods will need particular attention in your revision, though.**

We look forward to receiving your revised manuscript.

Kind regards,

Johannes Hönekopp

Academic Editor

PLOS ONE

Journal Requirements:

"This research was supported by funding from the Bill and Melinda Gates Foundation and Echidna Giving. "

5. Thank you for uploading your study's underlying data set. Unfortunately, the repository you have noted in your Data Availability statement does not qualify as an acceptable data repository according to PLOS's standards.

6. Please amend your list of authors on the manuscript to ensure that each author is linked to an affiliation. Authors’ affiliations should reflect the institution where the work was done (if authors moved subsequently, you can also list the new affiliation stating “current affiliation:….” as necessary).

7. We note that [Figure 1] in your submission contain [map/satellite] images which may be copyrighted. All PLOS content is published under the Creative Commons Attribution License (CC BY 4.0), which means that the manuscript, images, and Supporting Information files will be freely available online, and any third party is permitted to access, download, copy, distribute, and use these materials in any way, even commercially, with proper attribution. For these reasons, we cannot publish previously copyrighted maps or satellite images created using proprietary data, such as Google software (Google Maps, Street View, and Earth). For more information, see our copyright guidelines: http://journals.plos.org/plosone/s/licenses-and-copyright.

Reviewers' comments:

Reviewer's Responses to Questions

**Comments to the Author**

1. Is the manuscript technically sound, and do the data support the conclusions?

Reviewer #1: Partly

Reviewer #2: Yes

2. Has the statistical analysis been performed appropriately and rigorously? 

Reviewer #1: Yes

Reviewer #2: Yes

3. Have the authors made all data underlying the findings in their manuscript fully available?

Reviewer #1: Yes

Reviewer #2: Yes

4. Is the manuscript presented in an intelligible fashion and written in standard English?

Reviewer #1: Yes

Reviewer #2: Yes

5. Review Comments to the Author

Reviewer #1: This is an incredibly interesting paper and I wholeheartedly support this type of work. My main criticisms of the work pertain to the authors’ use of text processing methods. I’ve roughly organized my comments in the order of which the paper is also organized.

The authors should re-examine possible mismatches between their methodological approaches and their end goal of measuring “stereotypes” in text. For example, one of the weaknesses of using older, widely-used lexicons such as the Valence, Arousal, and Dominance (VAD) lexicon to measure stereotypes is that they were not originally intended for stereotype measurement. That is, how does valence or arousal actually relate to gender stereotypes? There is a large amount of psychological theory around what a stereotype is and how they may be operationalized, and also growing work in text processing that centers theory-driven frameworks. I suggest the authors look towards approaches outlined in “Theory-Grounded Measurement of U.S. Social Stereotypes in English Language Models” by Cao et al. 2022 in NAACL and “Understanding and Countering Stereotypes: A Computational Approach to the Stereotype Content Model” by Fraser et al. 2021 in ACL. I understand that many of the approaches in this paper borrow methods from work such as Adukia et al. 2023 and Lucy et al. 2020, which tend to have a “everything and the kitchen sink” methodological approach to analyzing school curriculum. For Lucy et al. 2020, this may have been fine because their main aim was to demonstrate the use of NLP (natural language processing) methods on textbook data, rather than the findings themselves. This current paper, as it recycles many tried-and-true text analysis approaches, may find that its more valuable contribution is around its findings, and so it should select its methods more carefully.

I am not sure why the authors employ both word co-occurrence and cosine similarity of word embeddings in this paper, as word embeddings are learned from word co-occurrences. What does one method offer that the other does not? In addition, at the end of section 3.3.2., the authors write “We calculate the similarity of embeddings as their cosine product.” What is a “cosine product”? Similarity of embeddings is usually calculated as cosine similarity, which is the dot product of two vectors divided by the product of their magnitudes.

Section 3.3.3: the section is titled as “part of speech tagging” but the description seems more like dependency parsing: “see how male and female characters are depicted, what actions they are depicted doing, and how active or passive they are (for instance depending on whether they are the subject or object of sentences)”. Please clarify whether this approach is actually part of speech tagging, or if it’s actually dependency parsing.

Section 3.2: “whether gendered words are capitalised or not, as an indication of whether characters are the active subject or passive object of a sentence”. This is a poor and noisy indication of whether characters are the active subject or passive object of a sentence. In many cases, the subject of a sentence might appear not as the first word in a sentence. Since the authors seem to be running dependency parsing already, why not use dependency parsed relations to determine whether someone is the subject or object? For example, the “nsubj” relation in UDPipe indicates that one word or token is the nominal subject of another.

Many of the results the authors show are over the entire corpus, but findings around how the provenance and subject area of data affect bias measurements are far more interesting than broad findings that reinforce the already-known consensus that textual data can reflect systemic gender biases. I suggest that the author lean less towards results and approaches that aggregate findings over the entire dataset. One of the strengths of this work is that it spans so many geographic regions, and I would like to see the findings emphasize this strength more. For example, how much of the variation we see in different measurements of gender bias is due to country of origin rather than subject matter of the textbooks, and vice versa?

Some more minor questions and notes:

What does it mean for textbooks to be “donor-funded”?

Section 2.1: “After selecting only files that were in English”; how did you do this?

Throughout the paper, some quotation marks are backwards, such as the starting quote around ‘names’ on the end of page 27. If this paper is typeset in LaTeX, the authors should consider using ` as starting quotes instead of ‘.

Does the released data that will accompany this paper include the OCR’d textbooks, urls of each source’s landing webpage, or downloaded pdfs’ source links? This could support the use of this dataset by future researchers.

The title of this article could be specified a bit more, e.g. ‘sexist textbooks’ is a bit vague and the title could highlight more of what makes this paper differ from the variety of other research that has quantified the existence of sexism in textual data.

Finally, much of the writing in this paper could be improved, and it could benefit from additional rounds of proofreading. For example, “New Analysis of image data” should be “New analyses of image data” on page 28 In addition, page 2 initially specifies 34 countries, but then this switches to 33 in the next paragraph.

Reviewer #2: The paper fulfills the requirements outlined by PLOS ONE: this represents original research and presents results not published elsewhere. The analyses are performed to a rigorous technical standard and are described in sufficient detail. The manuscript is presented in an intelligible fashion and is written in standard English. I appreciate the focus on books from a global sample so that we can further understand general patterns but also who might be exemplars in terms of gender representation. The authors have been very thoughtful in their analyses. This study presents a nice addition to the literature.

1. The description of the methods are clear and accessible to a layperson. This is a very exciting toolkit. The authors have taken great care in ensuring that readers can understand the work.

2. For the gender representation analysis, it would be interesting to see what the patterns are when including names for which they can identify gender. This could be analysis that goes in an appendix with the caveats of the precision and recall rates. Similarly, the occupation analysis is likely understating gender skew because of the exclusion of words such as salesman and actress.

3. The authors group books by subject into 14 groups, including “French” distinct from “Language” but not English (though in Figure 2, French is not separated). It is unclear how these groupings were made and why they are separated this way, especially across different countries. If the books are all written in the English language, how can other language books be included?

4. For the analysis using the valence, arousal, dominance lexicon (Table 3), would it make sense to have a similar analysis as to what was done in Table 4 in addition to showing the gender parity controls without log GDP?

5. How should we think about the generalizability of these results? The analysis draws on a broad set of contexts, but they all use English language. How do places that use English medium in their textbooks differ from places that use other languages?

6. A minor note for the writing: avoid using contractions such as “don’t” and “aren’t”

6. PLOS authors have the option to publish the peer review history of their article (what does this mean?). If published, this will include your full peer review and any attached files.

Reviewer #1: No

Reviewer #2: No

---

## [Author Response · Author response to Decision Letter 0]

24 Apr 2024

Thank you for the excellent comments, which have allowed us to substantially improve the paper. We detail our responses in full in the attached letter.

---

## [Decision Letter · Decision Letter 1]

23 Jul 2024

PONE-D-24-04364R1Sexist Textbooks: Automated analysis of gender bias in 1,255 books from 34 countriesPLOS ONE

Dear Dr. Crawfurd,

Thank you for resubmitting your paper "Sexist Textbooks: Automated analysis of gender bias in 1,255 books from 34 countries" to PLOS ONE.  Let me start by offering my apologies for the time it’s taken to get a decision back to you.  Your manuscript was originally assigned to another editor, and then was reassigned to me when they became unavailable.  

The resubmission was evaluated by the same two reviewers of the initial submission, and I have read it myself.  We all agree that the paper is much improved, and well on its way toward being able to be published.  

Both reviewers have provided brief comments for your consideration.  I’ll add that I agree with Reviewer 1 regarding their comment about dependency parsing not being an alternative to word embeddings.  Further, I thought the discussion at the top of page 10 could use some elaboration; for instance, by clarifying that the ‘relationships between’ words are structural, unlike word embeddings.  Perhaps a little more detail could be added about what dependency parsing is, and how it differs from the other two methods in the type of information it recovers.   Also, as a very minor comment, I noticed that the plural used for ‘corpus’ is ‘corpuses’ on line 460 and ‘corpora’ on line 569. 

Because accepted papers at PLOS One go straight to production, I'm taking the action of recommending Minor Revision for the manuscript, to give you the opportunity to address these comments to the extent you find appropriate, and to make any other minor changes as you see fit.  In taking this action, I do not intend to send a revised manuscript out for further external review, and anticipate accepting the paper on the next round. 

We look forward to receiving your revised manuscript.

Kind regards,

Andrew Kehler, Ph.D

Academic Editor

PLOS ONE

Journal Requirements:

Reviewers' comments:

Reviewer's Responses to Questions

**Comments to the Author**

1. If the authors have adequately addressed your comments raised in a previous round of review and you feel that this manuscript is now acceptable for publication, you may indicate that here to bypass the “Comments to the Author” section, enter your conflict of interest statement in the “Confidential to Editor” section, and submit your "Accept" recommendation.

Reviewer #1: (No Response)

Reviewer #2: All comments have been addressed

2. Is the manuscript technically sound, and do the data support the conclusions?

Reviewer #1: Yes

Reviewer #2: Yes

3. Has the statistical analysis been performed appropriately and rigorously? 

Reviewer #1: Yes

Reviewer #2: Yes

4. Have the authors made all data underlying the findings in their manuscript fully available?

Reviewer #1: Yes

Reviewer #2: Yes

5. Is the manuscript presented in an intelligible fashion and written in standard English?

Reviewer #1: Yes

Reviewer #2: Yes

6. Review Comments to the Author

Reviewer #1: This paper has improved from its original version; thank you.

One main comment for the authors to reflect on further: is it necessary to binarize how you define gender throughout the paper? For example, your name gendering process and exclude names that are ambiguously gendered. I suggest engaging with literature on how gender can be operationalized via text, and the limitations and caveats of how this operationalization could be done (e.g. Cao & Daumé “Toward Gender-Inclusive Coreference Resolution”). “Gender” can be expressed and perceived in a variety of ways, and different operationalizations of gender thus measure different types of gender. Precisely outlining your intentions and definitions of these social constructs is important in bias-related text-as-data work. Otherwise, one risks perpetuating other biases (e.g. exclusion of people whose names may not conform with some types of gender, such as their self-identified gender, or erasure of binary people). Similarly, for age, why must age be analyzed as a binary rather than on the continuum in which it naturally exists? I understand that for some reporting of results (e.g. word co-occurrence), binarizing measurements of social identity can simplify the reporting of results. However, for some findings, such as those related to representation, one could present distributions of names’ gender connotations.

With regards to discarding “salesman” and “actress” and possibly misestimating gender skew of occupations, one way of going about this would be to replace all instances of explicitly gendered occupations with a special token or gender-neutral form before conducting textual analyses. For example, replace all instances of “salesman”, “salesperson”, and “saleswoman” with “salesperson” in texts and then compute the association of gender with the token “salesperson” representing one occupation.

Line 328: “An alternative approach to word embeddings is dependency parsing”. These two methods are not alternatives to each other; they each measure very different things and should yield distinct results. I suggest revising the wording here.

Thank you for providing your data as well!

Reviewer #2: Thank you for addressing my comments. My only remaining comment is that you note that you "conservatively discard explicitly gendered occupations such as salesman or actress, which means that if anything we may understate the gender skew in occupations." I suggest adding an appendix exhibit that includes any gendered term (including salesman and actress) to highlight one of the bounds on the gender representation. These contribute to gender inclusion/exclusion.

7. PLOS authors have the option to publish the peer review history of their article (what does this mean?). If published, this will include your full peer review and any attached files.

Reviewer #1: No

Reviewer #2: No

---

## [Author Response · Author response to Decision Letter 1]

22 Aug 2024

We have responded in the attached letter

---

## [Editor Report · Decision Letter 2]

30 Aug 2024

Sexist Textbooks: Automated analysis of gender bias in 1,255 books from 34 countries

PONE-D-24-04364R2

Dear Dr. Crawfurd,

We’re pleased to inform you that your manuscript has been judged scientifically suitable for publication and will be formally accepted for publication once it meets all outstanding technical requirements.

Kind regards,

Andrew Kehler, Ph.D

Academic Editor

PLOS ONE
---

## [Editor Report · Acceptance letter]

6 Sep 2024

PONE-D-24-04364R2 

PLOS ONE

Dear Dr. Crawfurd, 

I'm pleased to inform you that your manuscript has been deemed suitable for publication in PLOS ONE. Congratulations! Your manuscript is now being handed over to our production team.

Kind regards, 

on behalf of

Dr. Andrew Kehler 

Academic Editor

PLOS ONE